# Elevated Homocysteine after Elevated Propionylcarnitine or Low Methionine in Newborn Screening Is Highly Predictive for Low Vitamin B12 and Holo-Transcobalamin Levels in Newborns

**DOI:** 10.3390/diagnostics10090626

**Published:** 2020-08-24

**Authors:** Tomaž Rozmarič, Goran Mitulović, Vassiliki Konstantopoulou, Bernadette Goeschl, Martina Huemer, Barbara Plecko, Johannes Spenger, Saskia B. Wortmann, Sabine Scholl-Bürgi, Daniela Karall, Susanne Greber-Platzer, Maximilian Zeyda

**Affiliations:** 1Austrian Newborn Screening, Division of Pediatric Pulmonology, Allergology and Endocrinology, Department of Pediatrics and Adolescent Medicine, Comprehensive Center for Pediatrics, Medical University of Vienna, 1090 Vienna, Austria; tomaz.rozmaric@meduniwien.ac.at (T.R.); vassiliki.konstantopoulou@meduniwien.ac.at (V.K.); bernadette.goeschl@meduniwien.ac.at (B.G.); susanne.greber-platzer@meduniwien.ac.at (S.G.-P.); 2Clinical Department of Laboratory Medicine, Medical University of Vienna, 1090 Vienna, Austria; goran.mitulovic@meduniwien.ac.at; 3Department of Paediatrics, Landeskrankenhaus Bregenz, 6900 Bregenz, Austria; martina.huemer@lkhb.at; 4Division of Metabolism and Children’s Research Center, University Children’s Hospital, 8032 Zürich, Switzerland; 5Department of Pediatrics and Adolescent Medicine, Division of General Pediatrics, University Childrens’ Hospital Graz, Medical University Graz, 8036 Graz, Austria; barbara.plecko@medunigraz.at; 6University Children’s Hospital, Paracelsus Medical University, 5020 Salzburg, Austria; j.spenger@salk.at (J.S.); s.wortmann@salk.at (S.B.W.); 7Department of Pediatrics I (Inherited Metabolic Disorders), Medical University of Innsbruck, 6020 Innsbruck, Austria; sabine.scholl-buergi@tirol-kliniken.at (S.S.-B.); daniela.karall@i-med.ac.at (D.K.)

**Keywords:** vitamin B12, newborn screening, homocysteine, methylmalonic acid, dried blood spots

## Abstract

Early diagnostics and treatment of vitamin B12 deficiency (B12D) in infants, mainly maternally conditioned, is crucial in preventing possible developmental delay and neurological deficits. Currently, B12D is rarely listed in regular newborn screening panels and mostly regarded as an incidental finding. The aim of this study was to evaluate a targeted newborn screening strategy for detection of suspected B12D. A decision strategy based on the primary parameters propionylcarnitine and methionine for selection of samples to be analyzed for total homocysteine by mass spectrometry was established. Therefore, 93,116 newborns were initially screened. Concentrations of vitamin B12 and holotranscobalamin in serum were obtained from clinical follow-up analyses of recalled newborns. Moreover, an extremely sensitive mass spectrometric method to quantify methylmalonic acid from the dried blood spots was developed. Overall, 0.15% of newborns were screened positive for suspected B12D, of which 64% had vitamin B12 concentrations below 148 pM. We also determined a cutoff value for methylmalonic acid in dried blood spots indicative for B12D in infants. Overall, we calculated a prevalence of 92/100,000 for suspected B12D in the Austrian newborns. In conclusion, we present a screening algorithm including second-tier measurement of total homocysteine that allows detection of low B12 serum concentrations with a high detection rate and low false-positive rate.

## 1. Introduction

Vitamin B12 (cobalamin; VitB12) deficiency (B12D) can have detrimental effects on the development of infants and is increasingly recognized as a global health problem [1,2]. B12D in newborns and breastfed infants is mainly a consequence of maternal deficiency caused by genetic polymorphisms, autoimmune diseases, gastrointestinal surgery, gastrointestinal infections, host-microbiota interaction, and low consumption of foods of animal origin. For the latter, especially vegans and vegetarians are vulnerable [1]. Severe B12D in infants leads to irritability, failure to thrive, apathy, anorexia, developmental delay, abnormal electroencephalography, and cerebral atrophy [2,3]. Severe B12D can easily and safely be treated either orally or with intramuscular administration of hydroxy-cobalamin, causing a rapid improvement in motor function and B12D related clinical symptoms [2,4]. Unrecognized and thus prolonged severe B12D bears the risk to develop irreversible neurological damage [5]. Though indicators of B12D can be detected via newborn screening, which may lead to avoidance of such risk [3,6], screening for B12D is rarely incorporated in newborn screenings and B12D is mostly regarded as a secondary condition or incidental finding [1,2].

VitB12 is a cofactor for enzymes catalyzing two intracellular metabolic reactions: remethylation of methionine from homocysteine (HCy) by methionine synthase in the cytosol, and conversion of methylmalonyl-CoA to succinyl-CoA by methylmalonyl-CoA mutase in the matrix of the mitochondria [1]. The remethylation pathway is crucial for nucleotide, myelin phospholipid, and neurotransmitter synthesis as well as DNA methylation. Therefore, B12D can cause genomic instabilities and neurological abnormalities [7,8].

The first consequence of inadequate VitB12 availability is the depletion of VitB12 stores in the liver in order to keep the amount of VitB12 bound to holotranscobalamin (holoTC), the active form of VitB12, constant in the blood. Exhaustion of the VitB12 stores results in the decrease of circulating holoTC, which in turn disrupts the two cellular VitB12 dependent reactions and results in accumulation of methylmalonic acid (MMA) and HCy [9]. Due to high interindividual variability, serum VitB12 but also holoTC concentrations are considered suboptimal diagnostic markers, while MMA and HCy levels give a better reflection of the cellular cobalamin metabolism status and thus indicate functional cobalamin deficiency [9].

Our aim was to develop, incorporate, and evaluate a sensitive and specific method for the detection of suspected B12D into the Austrian Newborn Screening Program. We here present an easily incorporable screening strategy for the screening for B12D in the newborn population with a comparably high detection rate. Furthermore, we propose a cutoff value for MMA in dried blood spots for detection of functional cobalamin deficiency in infants.

## 2. Materials and Methods

### 2.1. Subjects

Analyzable dried blood spot cards (*n* = 93,116) from newborns sent to the Austrian Newborn Screening between June 2018 and October 2019 were included in the study. This study was approved by the Ethics Committee of the Medical University of Vienna (EK1215/2016).

### 2.2. Primary Screening—Amino Acids and Acylcarnitines from Dried Blood Spots

Amino acids and acylcarnitines were determined from dried blood spot cards using a commercial CE-IVD Kit (Recipe Chemicals, Munich, Germany) on a binary HPLC pump-coupled TQD-Detector (Waters, Milford, MA, USA). from June 2018 till February 2019 (observation period 1, *n* = 59,557). After a system switch in the newborn screening program, from April 2019 until October 2019 (observation period 2, *n* = 33,559), amino acids and acylcarnitines were determined using a CE-IVD non-derivatized kit (Chromsystems, Gräfeling, Germany) on a Renata XEVO-TQD screening system (Waters, Milford, MA, USA).

### 2.3. Selection for Second Tier Analysis

The decision algorithm for cobalamin deficiencies was built on the available published data from newborn screenings [3,10,11] and own experience while taking the throughput capabilities of our facility into consideration. In detail, samples for total HCy (tHCy) measurement were selected by elevated C3 and its ratios to C2, palmitoylcarnitine (C16), methionine, and free carnitine (C0) and lowered methionine or its ratio to phenylalanine (Phe). Figure 1 shows the percentiles and absolute values (in µM) for the different cutoff values used for the two observation periods. Of note, referring to a recent publication [12], in the second period the selection scheme was modified to select more cases via the methionine- and less via the C3-dependent criteria with an emphasis on the C3/Met ratio (Figure 1).

### 2.4. tHCy from Dried Blood Spots

HCy concentrations are determined as total HCy (tHCy) i.e., the sum of reduced and non-reduced forms. After several optimization steps, the following protocol was established and validated.

Dried blood spot calibrators, quality control (Recipe Chemicals, Munich, Germany), and samples were punched using a 3.2 mm using the DBS Puncher^®^ Instrument (PerkinElmer, Waltham, MA, USA) into a 96-well PCR-clean microplate (Eppendorf, Hamburg, Germany). Internal standard was added followed by 15 µL reagent A (both Recipe Chemicals, Munich, Germany) and plates were incubated for 5 min at room temperature. 100 µL 0.2% HiPerSolv CHROMANORM^®^ for LC-MS formic acid (VWR, Lutterworth, UK) solution was added, and plates incubated for 10 min on a Titramax 1000 plate mixer (Heidolph Instruments, Schwabach, Germany). After 10 min centrifugation at 3000× *g* in a Centrifuge 5810 (Eppendorf, Hamburg, Germany), 2 µL of the supernatants were used for the analysis.

The separation and analysis were performed using a LC-MS/MS system (UHPLC Vanquish Flex-TSQ Quantis, ThermoFisher Scientific, Waltham, MA, USA) equipped with the Acquity UPLC BEH HILIC, 50 cm × 2.1 mm, 1.7 µm column (Waters, Milford, MA, USA) for separation. The mobile phase A consisted of 50% OPTIMA LC/MS Grade acetonitrile (Fisher scientific, Waltham, MA, USA), 10 mM CHROMASOLV™ LC-MS Ultra ammonium formate (Honeywell, Charlotte, NC, USA), 0.2% formic acid; mobile phase B consisted of 95% acetonitrile, 10 mM ammonium formate, 0.2% formic acid and the needle wash consisted of 90% acetonitrile. All dilutions were done using the LiChrosolv^®^ water (Merck, Dermstadt, Germany). The column-oven temperature was set at 30 °C and a flow rate was set to 650 µL/min. The separation gradient used was as follows: 0 min—89% B, 0.407 min—89% B, 0.732 min—30% B, 0.740—0.1% B, 1.058 min—0.1% B, 1.066 min—89% B, 2 min—89% B.

MS/MS settings for positive ionization mode were set as follows: spray voltage 1000 V; sheath gas: 45, auxiliary gas: 20, sweep gas: 1, ion transfer tube: 180 °C, vaporizer temperature: 515 °C, source fragmentation: 15 V and Collision-induced dissociation (CID) gas: 2 mTorr. For selected reaction monitoring (SRM), following *m/z* transitions for homocysteine were monitored: 136.1 *m/z* to 56.1 *m/z* and 136.1 *m/z* to 90.1 *m/z*; for homocysteine-d4 the monitored transitions were 140.1 *m/z* to 60.1 *m/z* and 140.1 *m/z* to 94.1 *m/z*. LC-MS data were analyzed using the TraceFinder™ v4.1 software (ThermoFisher Scientific, Waltham, MA, USA).

Homocysteine eluted at 1.05 min, fully separated from other analytes. The limit of quantification was determined to be at 6.4 µM using the equation: 10 × standard deviation of residuals/slope of the linear model. Repeatability within a single run was sufficient and it was determined to be: 9 µM—3.7% relative standard deviation (RSD), 17.3 µM—5.83% RSD and 36.5 µM—6.44% RSD. Intermediate precision over 1 month was 8% RSD at 17.3 µM and 8% RSD at 36.5 µM. After tHCy measurement, dried blood spots were stored at −80 °C for analysis of MMA.

### 2.5. Determination of Screening-Positives and Confirmation

For newborns with tHCy concentration over 10 µM (corresponding to the 89th percentile of the normal newborn population) a second dried blood spot card was requested. Patients were regarded screening-positive if tHCy was elevated for the second time or after a tHCy above 12 µM (corresponding to 96.7th percentile) from the first card was measured (Figure 1). Consequently, parents were requested to seek confirming diagnosis including determination of VitB12 and holoTC. Independently of that procedure, patients with C3 levels above alarm levels suggesting organic acidurias (not subject of this study) were directly contacted before tHCy analyses. Reporting the result of clinical workup in recalled screening positive newborns is part of the quality insurance of the Austrian Newborn Screening Program. However, these reports are not mandatory and thus data on patients’ confirmatory diagnostic workup are incomplete. Overall, in 42 of 77 and in 17 of 54 recalls, VitB12 concentrations and in 31 of 77 and 12 of 54 holoTC concentrations were obtained for the first and second period, respectively, at a median age of 4.5 weeks.

### 2.6. MMA from Dried Blood Spots

Calibrators and quality control standards were prepared at different concentrations by mixing known amounts of MMA (Merck, Dermstadt, Germany) with blood of a healthy donor which were spotted on filter cards.

Calibrators, quality control, and patient samples were punched into 96-well micro-plates (Eppendorf, Hamburg, Germany), deuterated MMA-d3 (Cambridge Isotope Laboratories, Tewksbury, MA, USA) as internal standard at a concentration of 0.64 µM in 0.4% acetic acid for LC-MS (Honeywell, Charlotte, NC, USA) was added and plates were centrifuged in a Centrifuge 5810 (Eppendorf, Hamburg, Germany) for 1 min at 3000× *g*, followed by a 15 min long incubation at room temperature on a Titramax 1000 plate mixer (Heidolph Instruments, Schwabach, Germany) at 1400 rpm. 100 µL of 0.4% acetic acid in CHROMASOLV™ LC-MS Ultra acetonitrile (Honeywell, Charlotte, NC, USA) was added and incubated for 10 min at room temperature followed by centrifugation (1 min at 3000× *g*). 70 µL of supernatants were transferred into a fresh 96-well microplate and mixed with 70 µL of acetonitrile and 8 µL of the processed sample were injected into the LC-MS/MS system (UHPLC Vanquish Flex-TSQ Quantis, ThermoFisher Scientific, Waltham, MA, USA).

The chromatographic separation has been performed using an InfinityLab Poroshell 120 HILIC Z; 2.1 × 50 mm × 2.7 µm (SPE), PEEK lined column (Agilent, Santa Clara, CA, USA). Mobile phase A consisted of 0.04% LiChropur™ ammonium hydroxide for LC-MS (Merck, Dermstadt, Germany), 5% CHROMASOLV™ LC-MS Ultra acetonitrile (Honeywell, Charlotte, NC, USA), and 100 mM CHROMASOLV™ LC-MS Ultra ammonium formate (Honeywell, Charlotte, NC, USA); mobile phase B consisted of 0.04% ammonium hydroxide, 85% acetonitrile and 30 mM ammonium formate. For dilution LiChrosolv Water for UHPLC-MS was used (Merck, Dermstadt, Germany). The column-oven temperature was set at 30 °C and the flow rate was set to 400 µL/min. The separation gradient was as follows: 0 min—99.9% B, 0.499 min—99.9% B, 0.6 min—93.7% B, 2.1—93.7% B, 4.6 min—55% B, 4.7 min—93.7% B, 5.9 min—93.7% B, 6 min—99.9% B and 6.5 min—99.9% B.

MS/MS settings for negative mode were as follow: spray voltage 250 V; sheath gas: 50, auxiliary gas: 10, sweep gas: 1, ion transfer tube: 180 °C, vaporizer temperature: 400 °C, source fragmentation: 16 V and Collision-induced dissociation gas: 1.5 mTorr. For SRM, following *m/z* transitions for MMA were monitored: 116.9 *m/z* to 73 *m/z* and for MMA-d3 the monitored transitions were: 119.9 *m/z* to 76 *m/z*. LC-MS data were analyzed using the TraceFinder™ 4.1 software (ThermoFisher Scientific, Waltham, MA, USA).

MMA was successfully separated from succinic acid with an elution time at 3.62 min. Furthermore, the MMA signal was reliably detected and quantified at 0.1 µM (4.2% RSD). Repeatability within a single run was sufficient: 2.2 µM—4.2% RSD, 9.3 µM—3.69% RSD. Intermediate precision over 1 month was 6% RSD at 2.2 µM.

### 2.7. Statistics

The obtained data were analyzed using R (version 3.6.2; https://www.r-project.org/) and RStudio (version 1.2.5001; https://rstudio.com/) software. Data were log-transformed to reach normal distributions allowing parametric statistics. Comparisons of two groups were performed with Student´s t-tests, for more than 2 groups ANOVA and Holm–Bonferroni posthoc tests were applied. Results were considered statistically significant when *p* < 0.05. The cutoff of MMA for B12D was determined by receiver operating characteristic curve (ROC) analysis for discrimination of newborns with known serum VitB12 concentrations below 148 pM [13] and elevated tHCy versus healthy newborns with normal tHCy and other screening parameters by the “closest top left” method.

## 3. Results

Measurement of MMA and tHCy, which are substrates of the two VitB12-dependent enzymatic reactions and thus biomarkers of functional B12D [14], require methods that are not feasible as first-tier analyses in newborn screening. For quantification of tHCy from dried blood spots; however, commercial kits are readily available. In contrast, propionylcarnitine (C3) and methionine, which are metabolites in these metabolic pathways upstream and downstream of MMA and HCy, respectively, and indicative for B12D, albeit with low specificity, are commonly determined in expanded newborn screening. Therefore, based on published data [3,10], we developed a decision algorithm based on C3 and methionine and distinct ratios of these markers to select samples for measurement of tHCy as a second-tier marker (Figure 1). Following a recent publication [12] and simultaneous with a change of the analytic instrument after 8 months of screening for suspected B12D, we adjusted the selection algorithm towards a higher number of samples selected via low methionine as well as a stronger emphasis on the C3/Met ratio (Figure 1).

Cutoffs were adjusted to keep the total number of samples to be selected for tHCy measurement close to 3%, a number defined by feasibility within our lab. Thus, 2.9% and 3.5% of all analyzed samples were selected for tHCy measurement in the first and second period, respectively (Figure 2A). The proportion of elevated tHCy remained essentially unchanged in both periods (4.4% and 4.6% of those analyzed for tHCy for period 1 and 2, respectively), resulting in 0.14% (77) and 0.17% (54) of screening-positive newborns with suspected B12D (Figure 2B).

We evaluated our screening performance with respect to the obtained serum VitB12 and holoTC concentrations. Both VitB12 and holoTC concentrations were low on average (114 pM and 116 pM serum VitB12 and 19.8 and 23.6 pM holoTC for period 1 and 2), without significant difference between the observation periods (Figure 3A,C). Applying the tentative cutoff values for B12D according to Green et al. [1] (<148 pM for VitB12, also referred to “acute deficiency” [14], <35 pM for HoloTC), the resulting positive predictive values were 69% (29 out of 42) and 59% (10 out of 17) based on serum VitB12 and 81% (25 out of 31) and 67% (8 out of 12) based on holoTC concentrations for period 1 and 2, respectively (Figure 3B,D).

To gain better insight into the performance of the developed B12D screening strategy, we quantified MMA, the most sensitive indicator of reduced functional cobalamin activity [14], from all screening-positive (*n* = 131) first dried blood spots. The mean values of MMA of screening-positive individuals were significantly elevated compared to newborns with normal primary screening results (mean 1.18 µM and 0.90 µM for period 1 and 2 vs. 0.21 µM, *p* < 0.001) and did not statistically differ from those with confirmed low VitB12 and holoTC concentrations (Figure 4A). We performed ROC analysis for MMA values to distinguish newborns with known serum VitB12 concentrations below 148 pM and elevated tHCy versus healthy newborns with normal tHCy and other screening parameters. The resulting area under the curve (AUC) was 0.96 (Figure 4B). Using the “closest top left” method, we determined the cutoff for MMA as an indicator of B12D to be 0.423 µM with 0.90 specificity and 0.91 sensitivity. Using this newly defined cut off, 87% and 82% of screening-positive samples showed elevated MMA values in both periods 1 and 2, respectively (Figure 4C).

Interestingly, the selection for tHCy analysis in the second period tended towards lower positive predictive values for low VitB12 and HoloTC and high MMA concentrations (Figure 3B,D and Figure 4C) compared to the first period. To gain information on the underlying cause, we analyzed the data with an emphasis on the selection pathways of the samples for the second-tier screening. Figure 5A shows that in the second period a markedly higher proportion of samples was selected via the Met pathway and, particularly pronounced, via the C3/Met ratio. However, MMA concentrations were markedly, though statistically not significantly (*p* = 0.15), lower in the samples selected via C3/Met in the second period (Figure 5B), meaning the lower cutoff resulted in a lower specificity, while there were no other apparent differences including the detection rate of newborns with low VitB12 concentrations (Figure 5C).

Finally, to benchmark our screening strategy we compared the overall estimated detection rate of low VitB12 concentrations in newborns to other published reports (Figure 6). Our estimated prevalence in the Austrian population of 93 in 100,000 newborns, which was 3–4 times higher compared to other reports [3,6,10].

## 4. Discussion

This study presents an easily incorporable screening strategy for low VitB12 concentrations in newborns with a significantly higher detection rate compared to previous studies and a low false-positive rate. Severe B12D can have detrimental effects on the development of the infants [2,5,15] and early, pre-symptomatic detection of severe deficiencies is regarded as advantageous [16,17]. Thus, the condition may well fulfill the criteria for neonatal screening programs as deduced from general screening criteria by Wilson and Jungner [18]. However, it is rarely screened for and cases are often regarded as incidental findings, differential diagnosis, or even as false-positives [19]. An explanation for this is that one of the Wilson and Jungner criteria, namely the consensus of whom and how to treat, has not yet been achieved. There is a lack of an exact definition of B12D due to the unreliability of blood biomarkers and missing information on the natural course. Since data in newborns and young infants are scarce, this problem is particularly pronounced for newborns.

Nonetheless, existing data underline that serum VitB12 levels below 148 pM and holoTC below 35 pM, as applied here as cutoffs, are well below the normal range [1,20,21]. Moreover, the high tHCy and MMA concentrations in our samples strongly indicate impaired intracellular VitB12 function. However, the actual odds and predictive factors to develop a detrimental outcome or to remain clinically asymptomatic with spontaneous remission of these biochemical abnormalities as well as the benefit of treatment need to be assessed.

Another reason for the underestimation of B12D is the understandable reluctance of parents and pediatricians to stress families and newborns by additional visits, blood sampling, and analyses for a condition that may not be regarded as a “real disease” but a condition that often normalizes without immediately detectable consequences. This bears the risk of unrecognized severe B12D that could lead to long-term detriments never attributed to B12D due to the low specificity of symptoms [5,22]. Here we show that applying a screening algorithm including only tHCy as a second-tier test that may be feasible for many newborn screening labs, newborns with low VitB12, low HoloTC, or elevated MMA can be identified with a positive predictive value between 59% and 87% (Figure 2 and Figure 3), which for newborn screening are remarkably high values [23]. The resulting low false positive rate of the screening algorithm shown in this study opens the field for focused research to determine the mid- and long-term impact of low VitB12 concentrations in the newborn age in order to develop an algorithm on whom and when to treat.

We show here that a cutoff of 0.42 µM (90th percentile) MMA in dried blood spots from newborns with elevated tHCy can predict VitB12 concentrations below 148 pM with around 80% positive predictive value. Thus, our data support that measuring MMA from already existing dried samples from newborn screening as a third-tier or confirmatory test may facilitate the diagnosis of B12D. Combination of MMA with tHCy may be an even better choice for a second-tier strategy to identify B12D. Analysis of MMA from dried blood spots, however, is not easily available for most newborn screening labs and a combined measurement of tHCy with MMA is technically even more demanding.

Although due to a lack of common definitions such comparisons are to be taken with care, compared to other reported estimates for prevalence of low VitB12, our estimate is 3–4 times higher [3,6,10]. These differences could be due to a different VitB12 status of different populations, but more probably a consequence of a different, improved screening algorithm. In two of these studies, individuals with low methionine were not considered for second-tier testing [3,10] and in a third study, a very high cutoff for MMA (2.35 µM/99.9th percentile) was applied [6]. Additionally, we learn from our data that neither C3-dependent parameters nor methionine-dependent parameters alone are sufficient for an effective first-tier selection since the overlap (i.e., samples with high C3 and low Met) is small (not shown). Additionally, the C3/methionine ratio alone is not sufficient. Lowering the cutoff for this parameter resulted mainly in a higher false-positive rate. In the future, Collaborative Laboratory Integrated Reports (CLIR) [24] or other machine learning-based tools [25] may further improve the specificity of B12D screening. Notably, also improvements in sensitivity may be possible since we do not know the real prevalence. Taking into account the high incidence of B12 deficiency in pregnant women as, for instance, shown in a Canadian study [26], the true prevalence of suboptimally VitB12-supplied newborns might be even higher than the number identified by the algorithm presented here. We hope to initiate more research on the situation of VitB12 supply of unborns and newborns. Though quantitatively playing a minor role—none of the newborns with known low VitB12 levels within our study population was reported to have a genetic cobalamin defect, also detection of such genetic defects would be improved by considering deficient VitB12 metabolism in newborn screening.

To conclude, by applying the presented screening strategy we find around 75 children per year in Austria (with about 85,000 births per year) with a potential risk of harm due to B12D. Further research is needed to assess the benefit of screening for B12D and consequent treatment of newborns. Finally, the high prevalence of putative B12D shown here indicates an unmet need for information concerning VitB12 requirement and supplementation of mothers during pregnancy and breastfeeding.

## Figures and Tables

**Figure 1 diagnostics-10-00626-f001:**
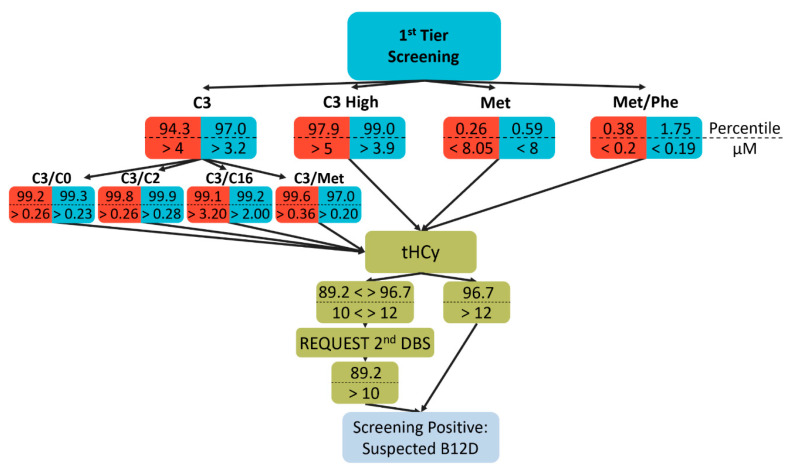
Screening decision algorithm for suspected vitamin B12 deficiency (B12D). Samples for total homocysteine (tHCy) measurement were selected by markedly elevated propionylcarnitine (C3 high) and in case of moderate elevation an additional elevation of one of the C3 ratios to, free carnitine (C0), acetylcarnitine (C2), palmitoylcarnitine (C16), and methionine (Met). Alternatively, lowered Met or its ratio to phenylalanine (Phe) leads to selection for tHCy measurement. Newborns were suspected for B12D if tHCY values were over 12 µM. tHCY measurement was repeated in a second dried blood spot if the primary value was between 10 µM and 12 µM. If the second measurement was also above 10 µM the newborn was categorized as screening positive. The values in the red and blue fields give percentiles (top) and absolute values (in µM, bottom) of used cutoff values for the first and second period, respectively. C3, propionylcarnitine; C3/C0, propionylcarnitine to free carnitine ratio; C3/C2, propionylcarnitine to acetylcarnitine ratio; C3/C16, propionylcarnitine to palmitoylcarnitine ratio; C3/Met, propionylcarnitine to methionine ratio; Met, methionine; Met/Phe, methionine to phenylalanine ratio.

**Figure 2 diagnostics-10-00626-f002:**
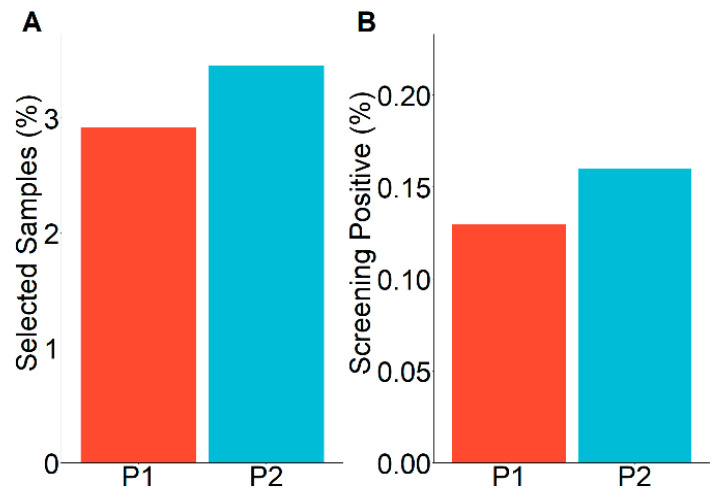
Screening overview. (**A**) Percentage of samples selected for second-tier measurement. Percentage of samples selected for total homocysteine (tHCy) measurement out of all samples analyzed according to the decision algorithm. Red bars show values for the first screening period, blue bars for period 2. (**B**) Percentage of screening positives. Percentage of screening-positive samples out of all samples analyzed according to the decision algorithm. Red bars show values for the first screening period, blue bars for period 2; tHCY, total homocysteine.

**Figure 3 diagnostics-10-00626-f003:**
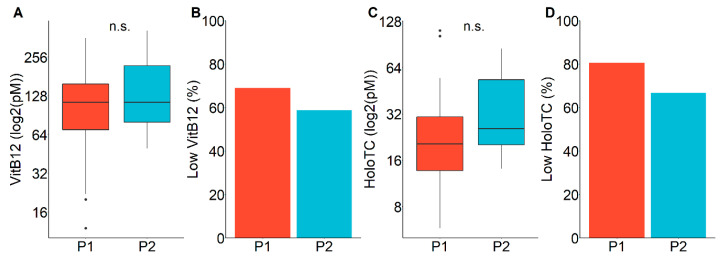
Evaluation of screening-positives by clinical laboratory parameters. (**A**) Serum vitamin B12 (VitB12) concentrations of screening-positives. *n* = 42 and 17, for period 1 and 2, respectively. (**B**) Positive predictive value for low VitB12. The graph shows the percentages of newborns being below the cutoff for low (<148 pM) out of 42 and 17, for period 1 and 2, respectively, screening positives with reported VitB12 concentrations. (**C**) Serum holotranscobalamin (holoTC) concentrations of screening-positives. *n* = 31 and 12, for period 1 and 2, respectively. (**D**) Positive predictive value of the screening algorithm for low holoTC. The graph shows the percentages of newborns being below the cut-off (<35 µM) out of =31 and 12, for period 1 and 2, respectively, screening-positives with reported concentrations. Red bars show values for the first screening period, blue bars for period 2; P1, first period; P2, second period of screening.

**Figure 4 diagnostics-10-00626-f004:**
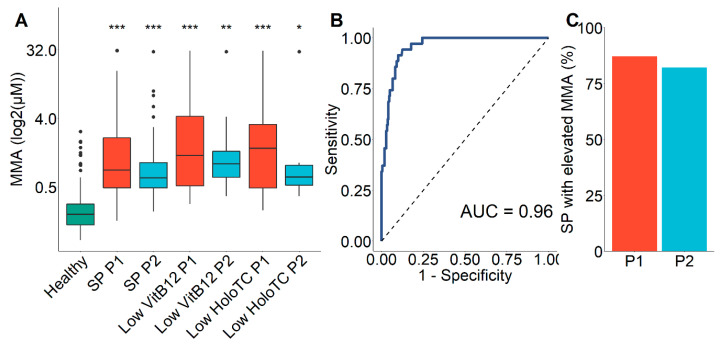
Evaluation of screening-positives by determination of, methylmalonic acid (MMA) from dried blood spots. (**A**) Markedly increased MMA concentrations in screening-positive samples. Boxplots for MMA serum concentrations of healthy newborn population (*n* = 242), screening-positives (*n* = 77 and *n* = 54 for period 1 and 2, respectively), screening-positives with known low serum vitamin B12 (VitB12) (*n* = 42 and *n* = 17 for period 1 and 2, respectively) and screening-positives with known low holotranscobalamin (holoTC) (N = 31 and N = 12 for period 1 and 2, respectively) concentrations are shown. *, *p* < 0.05; **, *p* < 0.01; ***, *p* < 0.001 vs. healthy, other comparisons were non-significant. (**B**) Receiver operating characteristic (ROC) curve analysis. Separation of newborns with normal and low VitB12 concentrations by MMA (cutoff: 0.423 µM). (**C**) Proportion of screening-positive samples with increased MMA concentrations. Red bars show values for the first screening period, blue bars for period 2 and green bars for healthy newborns; P1, first period; P2, second period of screening; SP, screening positive; AUC, area under the curve.

**Figure 5 diagnostics-10-00626-f005:**
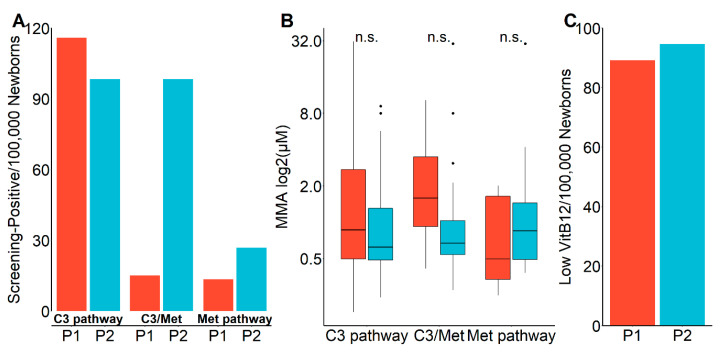
Impact of selection criteria on vitamin B12 (VitB12) screening performance. (**A**) Frequencies of screening-positives separated for the pathways via which samples have been selected for second-tier testing. Frequencies of samples selected via C3-dependent pathways (C3 high and C3 plus C3/C0, C3C2, or C3/C16), C3/Met, and methionine-dependent pathway (Met and Met/Phe), see Figure 1 for explanation, are depicted for indicated periods. (**B**) Methylmalonic acid (MMA) concentrations of samples selected via indicated pathways. Boxplots for MMA concentrations determined from dried blood spots of samples derived via indicated selection criteria and period. (**C**) Estimated frequency of low VitB12 in Austrian newborns. Frequency was estimated according to the number of screening-positives and proportion of low VitB12 concentrations within screening-positives. Red bars show values for the first screening period, blue bars for period 2; P1, first period; P2, second period of screening; n.s., non-significant.

**Figure 6 diagnostics-10-00626-f006:**
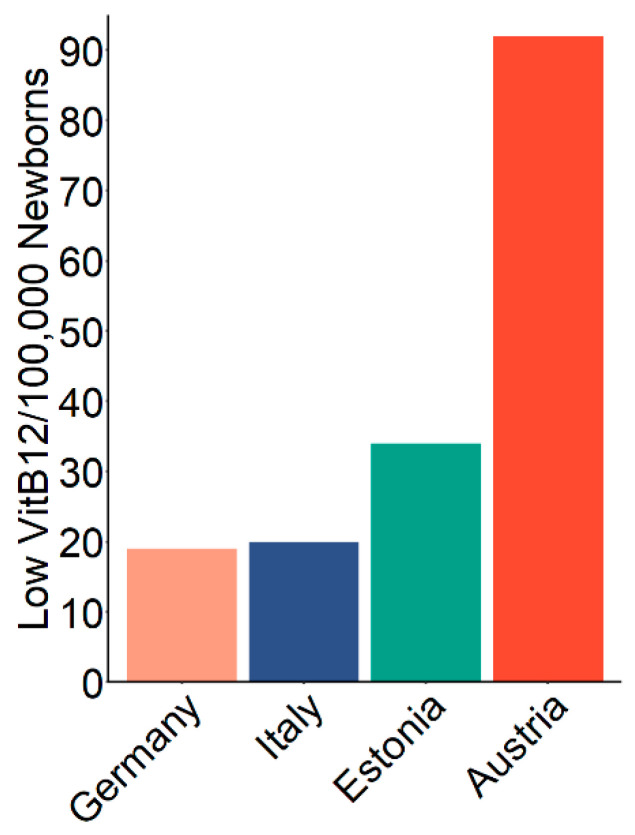
Estimated/reported detection of low vitamin B12 (VitB12) prevalence from different newborn-screening facilities. *Germany refers to reference* [6], *Italy to reference* [3] *and Estonia to reference* [10]. Beige bars show values for the estimated detection of low VitB12 in Germany, dark blue bars for Italy, green bars for Estonia and red bars for Austria.

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
