# Peer review of "Elevated Homocysteine after Elevated Propionylcarnitine or Low Methionine in Newborn Screening Is Highly Predictive for Low Vitamin B12 and Holo-Transcobalamin Levels in Newborns"

_diagnostics, 2020, doi:10.3390/diagnostics10090626_

Round 1

Reviewer 1 Report

minor revision:

row 95 change "total carnitine" with free carnitine

row 145 move "eppendorf" to line line 144 after micro plates

Add  row 204, 205, 206 and 207  to the figure 1 legend

Author Response

Response to Reviewer 1 Comments

Point 1: row 95 change "total carnitine" with free carnitine

Point 2: row 145 move "eppendorf" to line line 144 after micro plates

Point 3: add row 204, 205, 206 and 207  to the figure 1 legend

Response: Thank you! We incorporated all suggested improvements.

Reviewer 2 Report

High quality manuscript

Author Response

Response to Reviewer 2 Comments

Point: High quality manuscript.

Response : Thank you very much for this appraisal!

Reviewer 3 Report

To the authors,

In their work, the authors describe an important addition to newborn screening, which is currently not implemented in many national screening programmes. This requires the establishment of 2nd tier screening, which may even make 3rd tier screening necessary. However, the submitted paper raises a few questions which the authors still have to work on.

Page 2 Chapter 2.2: The authors determine the amino acids and acylcarnitines with the commercial kits from Recipe and Chromsystems. Both kits determine d4-DC + C5-OH as sum parameter, which cannot be separated in the non-butylated method. It would be interesting to know whether the authors observed an increased C4-DC + C5-OH in cases of methylmalonaciduria or in children with vitamin B12 deficiency. The parameter C4-DC increases moderately in the presence of methylmalonaciduria, but may still be in the upper range of normal values below the cut off.

Page 3 Line 95: To my knowledge, total carnitine is not determined in routine screening, only free carnitine. The authors do not indicate what you require the total carnitine for and how it is determined. The total carnitine can only be determined after alkaline or acid hydrolysis of the acyl carnitines.

Page 3 points 2.4 and 2.5: The authors use two chromatographic separations for the determination of homocysteine and methylmalonic acid. The reason for this is unclear; combined determinations have been described in the literature, which are also covered and accompanied by e.g. proficiency tests of the CDC.

Page 5 Line 195, Figure 1: The authors should describe how the criteria below 1st screening alone or in combination lead to 2nd screening. In addition, it would be desirable if absolute values or ranges of parameters were given in addition to the percentiles in the article.

Furthermore, the authors do not conclusively state how elevated propionylcarnitine is differentiated as propionaciduria, methylmalonaciduria or disorder of vitamin B12 metabolism. The first two diseases should be reported quickly to the submitting physician, a 2nd tier screening would lead to a delay in reporting the suspected disease.

The authors of the article do not address the fact that a vitamin B12 deficiency on the mother's side can be the cause and how many of the cases found are based on a maternal deficiency. Furthermore, it would be desirable that the underlying disorders (e.g. enzymatics, genetics) of the cases found be identified and reported.

Surprisingly, the authors found significantly more cases than have been described in the cited publications (Figure 5). There is no conclusive explanation for this. It would be helpful if the authors would compare and discuss their measured concentration ranges and cut-off values with those of the other publications.

It would also be interesting to know whether the authors have used the "Collaborative Laboratory Integrated Reports - CLIR" for the evaluation of their data and what supplementary information has been obtained from it.
